# Pre-Exercise Ingestion of Hydrogen-Rich Cold Water Enhances Endurance Performance and Lactate Response in Heat

**DOI:** 10.3390/medicina61071173

**Published:** 2025-06-28

**Authors:** Mariem Khlifi, Nidhal Jebabli, Nejmeddine Ouerghi, Fatma Hilal Yagin, Ashit Kumar Dutta, Reem Alwhaibi, Anissa Bouassida

**Affiliations:** 1Higher Institute of Sports and Physical Education of Kef, University of Jendouba, Kef 7100, Tunisia; khlifimaryem12@gmail.com; 2Research Unit: “Sport Sciences, Health and Movement” (UR22JS01), Higher Institute of Sports and Physical Education of Kef, University of Jendouba, Kef 7100, Tunisia; najm_ouerghi@hotmail.com (N.O.); bouassida_anissa@yahoo.fr (A.B.); 3Laboratory of Biochemistry, Faculty of Medicine of Tunis, Rabta Hospital, University of Tunis El Manar, LR99ES11, Tunis 1007, Tunisia; 4High Institute of Sport and Physical Education of Gafsa, University of Gafsa, Gafsa 2100, Tunisia; 5Department of Biostatistics, Faculty of Medicine, Malatya Turgut Ozal University, 44210 Malatya, Turkey; 6Department of Computer Science and Information Systems, College of Applied Sciences, AlMaarefa University, Ad Diriyah, Riyadh 13713, Saudi Arabia; adotta@um.edu.sa; 7Department of Rehabilitation Sciences, College of Health and Rehabilitation Sciences, Princess Nourah bint Abdulrahman University, Riyadh 11671, Saudi Arabia; rmalwhaibi@pnu.edu.sa

**Keywords:** pre-ingestion, molecular hydrogen, superficial cooling, lower limb muscles

## Abstract

*Background and Objectives:* Hyperthermia significantly limits endurance performance in hot environments. To enhance heat loss and optimize athletic performance, pre-cooling interventions can be employed to accelerate body cooling. Therefore, the aim of this study was to evaluate the effects of an internal pre-cooling intervention combined with external pre-cooling or hydrogen-rich water on endurance performance in the heat. *Materials and Methods:* In a double-blind crossover with counterbalanced trials, all participants underwent a shuttle run test after 30 min under the following conditions: (1) hydrogen-rich cold water ingestion (HRCW); (2) cold water ingestion and external pre-cooling (IEPC); and (3) cold-water ingestion (control). Maximal aerobic speed (MAS), number of shuttle run repetitions, dehydration, temperature, heart rate (HR), rate of perceived exertion (RPE), blood lactate, and feeling scale (FS) were measured during the 20 m shuttle run test. *Results:* Our results revealed a significant variation in dehydration, MAS, number of shuttle run repetitions, blood lactate, RPE, and FS (*p* = [0.001–0.036]); additionally, a significant group × time interaction was found for body temperature (*p* = 0.021). Post hoc tests revealed a significant change for MAS (HRCW: *p* < 0.001), number of shuttle run repetitions (HRCW: *p* < 0.001), dehydration (HRCW: *p*= 0.009; IEPC: *p* = 0.008), blood lactate (HRCW: *p* < 0.001; IEPC: *p* < 0.001), RPE (HRCW: *p* = 0.05; IEPC: *p* = 0.004), and FS (HRCW: *p* = 0.05; IEPC: *p* = 0.004), as well as a significant decrease in body temperature (IEPC: *p* < 0.001; HRCW: *p* = 0.028) compared to the control condition after the test. However, no significant differences were reported in HR among the different conditions. *Conclusions:* In conclusion, findings from this study suggest that ingesting hydrogen-rich cold water effectively mitigates the effects of heat stress, thereby improving endurance performance, enhancing mood, and reducing ratings of perceived exertion.

## 1. Introduction

It is well documented that sports competitions in hot environments demand higher thermoregulatory stress compared to sports competitions held in either normal or cold conditions [1,2]. Thus, the body temperature increases as an outcome of exercise-induced heat dissipation by athletes [3], and this consequently impairs physical performance [4]. In fact, the mechanisms of reduced physical performance in the heat are multifactorial, involving increases in thermal perception, cardiovascular strain [5], voluntary reductions in power output [6], and premature fatigue [7].

Cooling strategies or devices can be practically used by athletes to prevent hyperthermia and improve physical performance in the heat [2,8]. Indeed, previous studies, including reviews and meta-analyses, have demonstrated that pre-cooling can effectively limit the reduction in physical performance in hot weather [9]. Pre-cooling is a practical strategy for athletes that can be easily applied during warm-up in order to maintain or enhance performance in the heat [10]. In fact, different internal (e.g., ingestion of cold fluids or ice) and external (e.g., cooling clothing, cold water immersion, ventilation) cooling methods have been researched in sports performance [9,11].

Previous studies have demonstrated that external pre-cooling methods such as cold garments, ice bags, gel packs, and water immersion [12] effectively reduce thermoregulatory strain (core temperature, skin temperature) and improve power output, total covered distance, and time to exhaustion, as well as reduce completion time in hot environments [13].

Additionally, other studies have reported that internal pre-cooling, which involves the ingestion of cold water (<10 °C) or ice, is the most effective approach for minimizing thermal strain and for improving endurance performance [11,14]. Theoretically, pre-cooling with cold water reduces body temperature before exercise to increase heat storage capacity during activity and prolong the duration of exercise before reaching a critical level of hyperthermia [15].

Xu et al. [16] reported that the combined use of external and internal pre-cooling could further reduce temperature increase, decrease sweat loss to delay dehydration, and improve cardiovascular strain during endurance exercise in the heat.

Besides cooling techniques, hydration methods can also influence an athlete’s performance such as the use of hydrogen-rich water [17]. In fact, molecular hydrogen (H_2_) is recognized not only for its role as an antioxidant but also for its potential roles as a signaling molecule, inflammatory mediator, and mitochondrial bioenergetics regulator [17,18,19]. Specifically, H_2_ can regulate oxidative stress through the selective scavenging of hydroxyl radicals without affecting beneficial signaling reactive oxygen species [20]. When hydrogen molecules scavenge harmful reactive species, they help maintain the cells’ redox balance, protect cell health, and support smooth, efficient physiological functioning during endurance [21], repeated sprint ability [22], and maximal isokinetic muscle strength performance [23] when ingested as hydrogen-rich water (HRW) 30 min before exercise.

Such mechanisms could be particularly relevant in hot environments, where oxidative and metabolic stress are exacerbated [17,21]. Despite these promising insights, little is known about how molecular hydrogen interacts with thermoregulatory demands during prolonged exertion in the heat.

Therefore, this study aims to enhance knowledge by examining whether the combination of hydrogen-rich cold water (HRCW) with internal/external cooling strategies produces additive or synergistic effects on endurance and psychophysiological responses.

Furthermore, most existing research on pre-cooling strategies has focused on either internal or external methods in isolation, and only a few studies have explored their combination [15,16]. Among those, inconsistencies remain regarding whether combined cooling leads to superior performance outcomes or merely thermoregulatory advantages.

To the best of our knowledge, there have been no investigations evaluating the impact of hydrogen-rich cold water on endurance performance in the heat.

In this framework, the effect of hydrogen-rich cold water on physical performance should be justified to reveal their potential benefits resulting from heat stress conditions. Accordingly, the aim of this study was to examine the effects of an internal pre-cooling intervention combined with external pre-cooling or HRCW during warm-up on aerobic performance, body temperature, heart rate, blood lactate, perceived exertion, and feeling scale in the heat.

## 2. Materials and Methods

### 2.1. Study Design

Each participant completed two familiarization sessions under standardized conditions (indoor sports hall with a temperature of 31–33 °C and humidity of 38–44%) over a period of one week before the start of the study in order to minimize learning effects. During these sessions, participants were informed in detail about the shuttle run test procedure (see below) and were familiarized with the RPE scale and the feeling scale. Additionally, anthropometric measurements were accomplished for the participants. These sessions ensured that participants were fully prepared for the experimental conditions required for the study.

During experimental sessions, a double-blind, counterbalanced crossover design was used, involving the three following conditions: (1) ingestion of 500 mL of hydrogen-rich cold water (HRCW) before warm-up; (2) ingestion of 500 mL of cold water and external pre-cooling of lower limb muscles (IEPC) before warm-up; (3) and ingestion of 500 mL of cold water (control).

### 2.2. Participant

The sample size of our study was determined using the power analysis program G*Power (version 3.1.9.3, University of Düsseldorf, Germany). An a priori power analysis was calculated using the F-test family (repeated one-way ANOVA measures) using our primary outcome at an assumed power of 0.95, an alpha level of 0.05, and a medium effect size of 0.33 for a sample size of 22 participants.

Accordingly, 22 healthy male sports science students volunteered to take part in this study (Table 1).

Inclusion criteria for participants in this study were as follows: all participants were sports science students, performing similar physical exercise volumes as part of their weekly university training (16 h wk^−1^) including different sports such as combat sports, basketball, football, and athletics. Also, all participants reside in the same region (North West Tunisia), which has acclimatized to average summer temperatures between 32 °C and 36 °C over the past four years. The exclusion criteria were having a disability or having experienced orthopedic injuries or surgeries that limited movement; having comorbidities (e.g., cardiorespiratory failure, asthma); and participating in any other training programs during the period of the intervention and having missed any of the testing sessions.

Participants were instructed to refrain from vigorous exercise within 48 h, maintain usual hydration, dietary habits, sleep patterns, and avoid ingesting ergogenic supplements (e.g., caffeine, vitamins, etc.) in the 24 h before each testing session.

After receiving a description of the protocol, risks, and benefits of the study, each volunteer provided written informed consent. All participants successfully completed this study (Figure 1). The local research ethics committee approved the study (18 January 2023; UR22JS01) in accordance with the Declaration of Helsinki. Written informed consent to participate in this study was provided by the participants.

### 2.3. Experimental Procedure

This study is a randomized, double-blind, crossover trial examining the impact of an internal pre-cooling intervention combined with external pre-cooling or HRCW during warm-up on physical performance and psychophysiological responses during a shuttle run test in the heat.

All the participants performed the three procedures, with two experimental procedures and one control, in a randomized crossover model (www.randomizer.org; 1 August 2023).

During each session, a 20 m shuttle run test was performed. A period of a one-week washout separated the sessions. Hydration was carried out 30 min before exercise during each session.

Before each session, participants performed a standardized 10 min warm-up, including light jogging (heart rate: 110–145 beats·min^−1^) and lateral displacements without any stretching exercise.

Sessions were carried out in an indoor sports hall (14 × 22 m) in hot weather (month: August 2023; indoor air temperature 32 ± 2 °C; humidity 40 ± 4%). The order in which the trials were performed was randomized and counterbalanced.

All tests were carried out at the same time of day (between 4:00 pm and 5:00 pm) to minimize the influence of the circadian rhythm on each protocol [24]. Figure 2 illustrates the experimental flow for the study.

#### 2.3.1. Protocols

##### Internal and External Pre-Cooling

The internal pre-cooling technique was implemented by ingesting 500 mL of water cooled to a temperature of 1–3 °C 30 min before testing under all conditions. The mineral water was stored in a thermal box to maintain temperature. The mineral water temperature was controlled using an infrared thermometer (UT301C, UNI-T, China). A cold water temperature of around 1–3 degrees was chosen based on previous studies [25,26].

For the external pre-cooling technique, a gel ice pack, a large pack that ensured superficial cooling of the muscles with a measurement of 38 cm in length and 14.5 cm in width, was used. During the 10 min cooling time, the participants were lying on a mat and cold pack compression was carried out by an investigator. The cooling pack was designed to keep cold compresses in close proximity to the surface of the skin of the lower limbs (anterior thigh; posterior thigh; anterior calf; posterior calf) in order to optimize pressure cooling. Due to their portability, the ice/cooling pack was used for all four zones by the investigator; this modality is more practical and faster than immersing the lower limbs in cold water before physical exercise. Hydration was not permitted during the cooling period. The two cooling techniques were employed, consistent with internal and external cooling strategies [12].

##### Hydrogen-Rich Cold Water

Participants underwent a double-blind study and received 500 mL of cold mineral water under either control or IEPC conditions (oxidation reduction potential: +167 mV; temperature: 1–3 °C) or hydrogen-rich cold water (oxidation reduction potential: −650 mV; temperature: 1–3 °C) using the same source of mineral water (pH: 7.4).

Under the HRCW condition, hydrogen concentration (H_2_) in the water was 1600 ppb, as reported by the manufacturer (Headrogen Health, bottled water, Hydrogen Health PTY LTD, Australia). This concentration was achieved by means of a 3 min electrolysis, whereby the apparatus decomposes water into oxygen and hydrogen gases. The chemical properties of the HRCW included were chosen similarly to previous studies [17,21]. All the 500 mL water had to be drank immediately after the hydrogenation process (30 min before test) in order to maintain hydrogen concentration levels and avoid a decrease in oxidation reduction potential. Both drinks, mineral cold water or HRCW, were served in visually identical plastic bottles.

Participants could not distinguish between HRCW and mineral cold water because H_2_ is colorless, odorless, and tasteless [17,21,27], it was not possible to distinguish HRW from cold mineral water using the human senses. Previous studies have identified that the procedure of H_2_ supplementation as described here meets certain H_2_ intake needs without causing side effects [28,29].

To ensure the double-blind trial design, all water was blinded and randomized by a separate researcher who was not involved in the data collection. Also, hydration was not allowed before and during the shuttle run test for all sessions.

#### 2.3.2. Measurements

##### Anthropometric Measurements and Percentage of Dehydration

Anthropometric measurements were performed with subjects bare-footed and lightly clothed. Height was measured in meters (m) using a wall-mounted stadiometer with a precision of (±0.1 cm). Body mass was measured in kilograms (kg) using bioelectrical impedance analysis (*Omron HBF-500*, Omron Healthcare Co., Ltd., Mannheim, Germany) with a precision of (±0.1 kg). Body mass index (BMI) was calculated as follows:BMI (kg/m^2^) = body mass (kg)/height (m)^2^.

Participants were weighed nude just before the test and again at the end of the test, after being dried, to measure the percentage of dehydration, which is equivalent to the amount of water lost through sweat, calculated as follows [30]:Dehydration (%) = [(Pre-body mass − Post-body mass)/Pre-body mass] × 100.

##### Shuttle Run Test

The shuttle run test started with a running speed of 8.5 km·h^−1^ and increased by 0.5 km·h^−1^ every minute until exhaustion between two lines 20 m apart, following the beep emitted by the beeper (Sport Beeper Pro, Best Electronic, Rhône-Alpes, France) [31]. Successful runs were defined as completed shuttles. The test was terminated if the participant failed to cross the line (within 2 m) at each beep on two consecutive occasions. Maximal aerobic speed (MAS) and the number of shuttle run repetitions were calculated at the last successfully completed stage.

The reproducibility of tests was determined using pilot data from 22 participants collected during familiarization trials (shuttle run test: intra-class correlation coefficient (ICC) = 0.990, 95% confidence intervals (CI): 0.981–0.994).

##### Infrared Tympanic Temperature and Ambient Temperature

An infrared tympanic membrane thermometer (BRAUN IRT-3020, Hessen, Germany) was used to measure the core temperature of both the right and left ears. The thermometer probe was inserted into the ear canal, ensuring that it fit snugly without forcing. It only took 2 s for each ear to measure the infrared tympanic temperature. The ear canal was cleaned and dried before each measurement according to the manufacturer’s recommendations. Each subject had their own hygienic cap attached to the thermometer at each session. No fluids were consumed prior to infrared tympanic temperature measurement in any of the test phases (pre-post-test) unless otherwise directed by the investigators in accordance with the study protocol. The mean of the two readings was recorded before and after the shuttle run test.

Infrared tympanic temperature was the best estimator of core temperature during the exercise in the heat [32], with an error range of ±0.286 °C and ±0.392 °C [33].

Under each condition, ambient indoor sports hall temperature was measured with a PT-100 hand-held thermometer (Electronic Temperature Instrument Ltd., West Sussex, UK). The accuracy of this thermometer is ±0.1 °C.

##### Heart Rate and Bood Lactate

During the shuttle run test, HRpeak and HRmean were recorded using an individual heart rate monitor (polar electro, TF4).

Three minutes after the test, blood lactate concentration was measured by taking a sample from the third fingertip using a portable lactate monitor (lactate Pro 2).

##### Feeling Scale

The feeling scale (FS) was measured to assess the participants’ mood after the shuttle run test. The FS is an 11-point single-item scale ranging from +5 (very good) to −5 (very bad) with a midpoint of 0 (neutral) [34]. The participants received standard instructions regarding the use of the FS after the shuttle run test, as it is quite common that mood changes during exercising. The instructions were as follows: “try to inform us, by a number, on your inner feeling during the test”.

##### Rating of Perceived Exertion

After the test, the rate of perceived exertion (RPE) was determined using Borg’s RPE (6–20) Scale [35].The 15-point scale ranged from 6 (“very, very light”) to 20 (“very, very hard”) and was used to verbally explain the RPE after the tests. Each player responded to the following question: “How did you perceive your exertion in your full-body during the test?”.

### 2.4. Statistical Analysis

All statistical tests were processed using SPSS IBM^®^ software v.22 (SPSS Inc., New York, NY, USA). Data were expressed as means ± standard deviations (SDs). Data normality and homogeneity were assessed through the Shapiro–Wilk test and all variables showed normal distribution. The reliability of all tests was assessed by the intra-class correlation coefficient (ICC) and 95% confidence intervals (95%CIs) [36].

The data of the shuttle run test (MAS, number of shuttle run repetitions, HR mean, HR peak, RPE, FS, and blood lactate) were examined using a repeated one-way ANOVA. The body temperature outcome was analyzed using a two-way (3 (conditions) × 2 (times)) ANOVA with repeated measures. When the ANOVA revealed a significant difference, a LSD adjusted post hoc test was used in the form of pair-wise comparisons. Partial eta squared (η^2^*_p_*) was calculated using 0.0099, 0.0588, and 0.1379 considered as small, medium, and large effect sizes, respectively, based on guidelines adapted from Cohen’s benchmarks for eta squared [37]. Also, Cohen’s d (d) was calculated with demarcations of trivial (<0.2), small (0.2–<0.50), moderate (0.5–<0.8), and large (≥0.8) used as effect sizes for all statistical variables [38]. In addition, the upper and lower 95% CIs of the difference (95% CIs) were calculated for the corresponding variation. Moreover, percentage changes between conditions in the MAS and the number of shuttle run repetitions have been added using the following formula: Δ (%) = ([condition x/condition y] − 1) × 100. The level of statistical significance was set to *p* < 0.05 [39].

## 3. Results

### 3.1. Physical Performance and Heart Rate

Overall, there was a significant difference in the MAS and the number of shuttle run repetitions between the condition (MAS: *p* = 0.009; η^2^*_p_* = 0.464; power = 0.857; number of shuttle run repetitions: *p* = 0.004; η^2^*_p_* = 0.447; power = 0.908). The post hoc test revealed a significant increase in the MAS, and the number of shuttle run repetition values were observed only during HRCW ingestion (MAS: *p* = 0.001, d = 0.33, Δ = 4.8%, 95% CI = 0.34 to 1.04; number of shuttle run repetitions: *p* = 0.001, d = 0.40, Δ = 20.38%, 95% CI = 8.69 to 28.65) compared to the control condition. However, no significant changes were recorded between the other conditions. In addition, no significant change was observed (*p* > 0.05) for HRpeak and HRmean between different conditions (Table 2).

### 3.2. Dehydration

Overall, there was a significant variation in dehydration between conditions (*p* = 0.016; η^2^*_p_* = 0.429; power = 0.799). The post hoc test revealed a significant reduction in dehydration for the HRCW condition (*p* = 0.009, d = 0.60; 95% CI = −0.28 to −0.045) and IEPC (*p* = 0.008, d = 0.71; 95% CI = −0.27 to −0.05) compared to the control condition (Table 2).

### 3.3. Blood Lactate

Regarding metabolic responses, there was a significant difference in blood lactate values between conditions (*p* < 0.001, η^2^*_p_* = 0.771; power = 1). The post hoc test revealed that a significantly lower blood lactate concentration post-exercise was recorded during HRCW ingestion (*p* < 0.001; d = 0.81; 95% CI = −0.84 to −0.39) and IEPC (*p* < 0.001; d = 0.79; 95% CI = −0.82 to −0.40) compared to the control condition. However, no significant difference was observed between HRCW and IEPC for blood lactate concentration (Table 2).

### 3.4. RPE and Feeling Scale

Overall, there was a significant difference between the RPE and FS values between conditions (RPE: *p* = 0.036; η^2^*_p_* = 0.370; power = 0.685; FS: *p* = 0.014; η^2^*_p_* = 0.439; power = 0.816) (Table 2).

The post hoc test revealed a significant reduction in RPE values reduced during internal pre-cooling with HRCW (*p* = 0.05; d = 0.59; 95% CI = −0.58 to −0.01) and IEPC (*p* = 0.004; d = 0.88; 95% CI = 0.17–0.79) compared to the control condition.

Moreover, the post hoc test revealed a significant increase in positive FS values during HRCW ingestion (*p* = 0.007; d = 1; 95% CI = 0.36 to 1.17) and IEPC (*p* = 0.012; d = 0.83; 95% CI = 0.27 to 1.92) compared to the control condition. However, there was no significant difference in FS values between HRCW and IEPC.

### 3.5. Body Temperature

The pre- and post-test results of body temperature are outlined in Figure 3. A significant main effect of time was observed for body temperature (*p* < 0.001; η^2^*_p_* = 0.973; power = 0.99). A significant group × time interaction was found for body temperature (*p* = 0.021; η^2^*_p_* = 0.109; power = 0.752).

Post hoc tests indicated a significant decrease in body temperature IEPC (*p* < 0.001; d = 1.27; 95% CI = −0.34 to −0.27) and HRCW (*p* = 0.028; d = 0.69; 95% CI = −0.23 to −0.20) compared to the control condition after the shuttle run test.

Contrast analysis revealed a significant decrease in body temperature during the IEPC condition compared to the control (*p* = 0.001) condition.

## 4. Discussion

To our knowledge, this is the first study to examine the effects of an internal pre-cooling intervention combined with external pre-cooling or HRCW on aerobic performance, dehydration, body temperature, heart rate, blood lactate, perceived exertion, and feeling scale during the shuttle run test under hot conditions. The main finding of the present study revealed that the MAS and the number of shuttle run repetitions were significantly higher only under the HRCW condition compared to the control condition. This improvement in aerobic performance is accompanied by a significant reduction in dehydration, blood lactate, and RPE. Additionally, there was a significant increase in the FS under the HRCW and IEPC conditions compared to the control condition. Moreover, a significant decrease in body temperature was observed during the IEPC and HRCW conditions compared to the control condition. However, no significant differences in HR were found across the different conditions. Previous studies have investigated the effects of each modality separately on aerobic performance [17,40].

The current results suggest that the pre-exercise ingestion of 500 mL of HRCW is the most effective strategy for improving the MAS. In other words, using HRCW as a pre-cooling strategy provides a performance benefit (MAS, d = 0.33) during the shuttle run test under heat stress.

To our knowledge, this is the first study to demonstrate the effect of pre-ingesting HRCW on physical performance in the heat. However, in agreement with our results, Xu et al. [21] reported that a dose of 500 mL of HRW taken 30 min before exercise at normal ambient temperature significantly enhanced the MAS during the shuttle run test. However, other studies have shown no significant effect of HRW at different doses on physical performance during maximal and submaximal exercises [41].

The improvement in the MAS can be attributed to HRW’s ability to enhance performance during exercise. This enhancement is linked to adaptive responses to physical stress which help maintain the balance between oxidants and antioxidants through enhanced mitochondrial biogenesis, improved antioxidant defenses, and increased muscular endurance [17].

Additionally, the significant impact of H_2_ on metabolism may partly explain these results, as H_2_ remains present in the lipid and glycogen phases for a longer duration [41,42]. This extended presence can enhance energy availability during prolonged exercise, reducing the metabolic strain that often accompanies thermal stress.

In addition, H_2_ may stimulate mitochondrial function, increasing ATP levels and stabilizing membrane potential, which are key contributors to endurance during heat stress [19,23,43]. H_2_ can react with aggressive oxidants such as hydroxyl radicals (OH). Consequently, these processes fine-tune calcium and mitochondrial ATP-sensitive K^+^ channels in a way that helps stimulate downstream ATP production [44]

Previous reports were concerned with minor effects or insignificant gains in time-to-exhaustion performance when either internal or external cooling were applied [13,15], but our results show a definite enhancement in performance when HRCW was used. This implies that hydrogen may synergize with the cold water’s effect of lowering core temperature rather than simply being additive.

Taken together, the results suggest that HRCW is more than a chilled drink. Molecules of hydrogen in the mix could team up with the cooling effect, combating both thermal and oxidative stress. While cold water mainly holds heat off by raising tissue storage space [15], the added H_2_ might also optimize metabolism and dial down neuromuscular fatigue through stronger mitochondria and lower lactate peaks [17,19].

HRCW raised the maximal aerobic speed by roughly 4.8% and increased shuttle run repetitions by more than 20%, figures that match or even surpass results seen with hydrogen-rich water in standard environments [17,21]. These gains suggest that HRCW can lessen heat-induced performance drops as much as or more than classic pre-cooling strategies.

The potential effect of HRW on improving aerobic performance may be limited in a warm environment, which requires further investigation. Exercising under such conditions triggers a more pronounced plasma antioxidant and anti-inflammatory response [43], highlighting the need for a deeper exploration of the effectiveness of HRW and internal cold water.

The positive effect of HRCW on reducing blood lactate levels after exercise could also explain our results. Aoki et al. [23] demonstrated that ingesting HRW before exercise reduced blood lactate levels and improved muscle function, as lactate serves as a key energy source for ATP resynthesis during prolonged exercise.

In the context of our results, it is unclear whether HRW reduced anaerobic lactate production

However, Dohi et al. [19] observed that the antifatigue effects of H_2_ may be mediated by improved metabolic coordination and immune redox balance, particularly through increased lactate dehydrogenase and glutathione peroxidase activity, which could also explain the improved performance. Moreover, hydrogen has the ability to block oxidants and has anti-inflammatory and anti-allergic effects [45]. Hydrogen plays a critical role in preventing oxidative damage and cellular injury; at the same time, it helps improve mitochondrial function without being cytotoxic to improve exercise performance [28]. These protective effects may also play an additional role in reducing heat stress, as mitochondrial function helps maintain cellular homeostasis in the heat. In addition to metabolic effects, H_2_ can decrease the sensation of muscle stiffness, at least after a simple strength exercise session and after downhill running [43,46].

It is logical to expect that internal and/or external pre-cooling may positively impact reducing body heat storage, as well as increasing heat storage capacity [47]. However, no study has evaluated the thermoregulatory effects of internal pre-cooling with HRW, unlike what has been observed with external pre-cooling [2,16].

Taken together, existing studies supports our findings, indicating that an internal pre-cooling strategy reduces internal or rectal temperature by 0.5 ± 0.1 °C prior to exercise, which improves thermal comfort [48]. This increased comfort can increase the skin’s evaporation potential and consequently delay the onset of thermal fatigue [49]. Additionally, cold water passing through the esophagus may reduce blood temperature in the carotid artery, potentially lowering brain temperature and enhancing performance in the heat [50]. However, in the present study, the combination of internal and external pre-cooling was not shown to produce an ergogenic strategy on physical performance, contrary to previous studies [15]. Roriz et al. [15] reported that the combination of external and internal cooling methods, involving drinking cold water and wearing a cold vest, produced a thermoregulatory effect, thus improving running duration at an intensity of 80% VO2_max_ in hot and humid conditions. They also reported that combined pre-cooling could decrease sweat loss to delay water loss during exercise in hot weather. Similarly, Tyler et al. [48] observed that internal pre-cooling may be even more impactful than external cooling on aerobic performance in hot weather.

Consequently, the internal and external pre-cooling of lower limbs does not improve endurance performance. In this sense, to further incorporate these findings with ours, we can indicate that internal and external pre-cooling of the body area with high tissue perfusion has proven to be beneficial in reducing body temperature and body loss percentage without changes in aerobic performance. These benefits may be lost during exercise due to external vasoconstriction or a significant decrease in muscle temperature before the start of the exercise [48].

Generally, during exercise in the heat, increased thermal stress and body temperature are associated with improved RPE values [47]. In this context, the pre-cooling used by HRCW and IEPC induced thermoregulatory benefits by reducing the mean RPE post-test compared to the control condition. The reduction in heat stress could be due to the decrease in the average body temperature observed during IEPC and HRCW ingestion, which could be a factor in the reduction in the RPE.

It makes sense that heat stress has a negative impact on mood and cognitive functions during exercise [51,52]. In this case, the application of HRCW and IEPC proves to be beneficial for the feelings of freshness and comfort for the participants during an exhaustive exercise in the heat.

Similarly, the sensations of pleasure experienced when ingesting cold drinks or localized external cooling generate powerful sensory effects that can stimulate pleasure centers in the brain [53].

To summarize, pre-cooling conditions combined with HRCW improve post-test mood states. However, these improvements do not necessarily reflect improved physical performance during aerobic exercises with progressively increasing loads. These results also require further investigation in future studies into the kinetics of the RPE and FS throughout exercise to better understand their variations.

Some limitations warrant consideration. First, the volume of HRCW ingested was constant across each subject and the dose was not standardized according to body mass. Second, there was an absence of oxidative stress markers, which could provide further information on the effect of HRCW on physiological responses. Third, the application of pre-cooling strategies requires the assessment of the intramuscular temperature of the lower limb by a needle microprobe during the test in order to study intramuscular changes during the test. Finally, the practical application of HRCW in women should be studied with great caution, given that there are sex differences in thermoregulatory responses and body composition [49].

As practical applications, the ingestion of 500 mL of hydrogen-rich cold water 30 min before warm-up could favorably influence physical, physiological, and perceptual responses and body temperature without wasting time in external pre-cooling applications.

## 5. Conclusions

Findings from this study suggest that ingesting hydrogen-rich cold water effectively mitigates the effects of heat stress, thereby improving endurance performance, enhancing mood, and reducing ratings of perceived exertion. However, internal pre-cooling combined with external pre-cooling may be equally effective in increasing mood state and reducing the post-test RPE with no effect on physical performance.

## Figures and Tables

**Figure 1 medicina-61-01173-f001:**
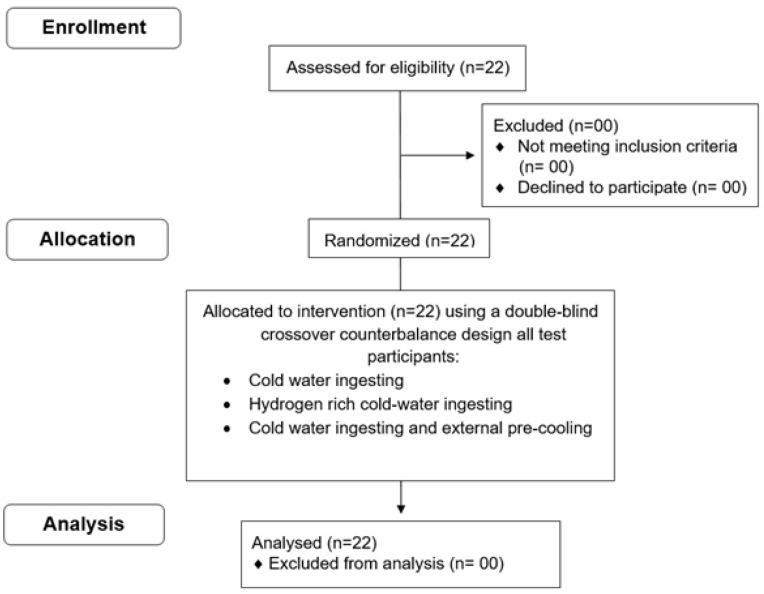
Flow diagram.

**Figure 2 medicina-61-01173-f002:**
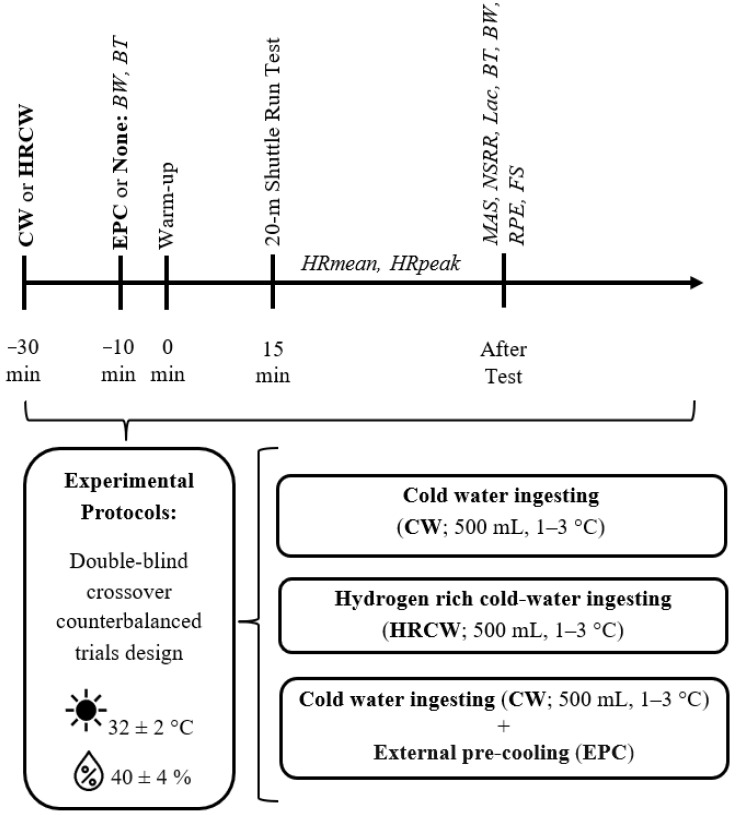
Flow process diagram of the entire experimental protocol, which includes the procedure and measurements. CW: cold water; BW: body weight; BT: body temperature; HRmean: heart rate mean; HRpeak: heart rate peak; MAS: maximal aerobic speed; NSRR: number of shuttle run repetitions; Lac: lactate, RPE: rate of perceived exertion; FS: feeling scale; HRCW: hydrogen-rich cold water.

**Figure 3 medicina-61-01173-f003:**
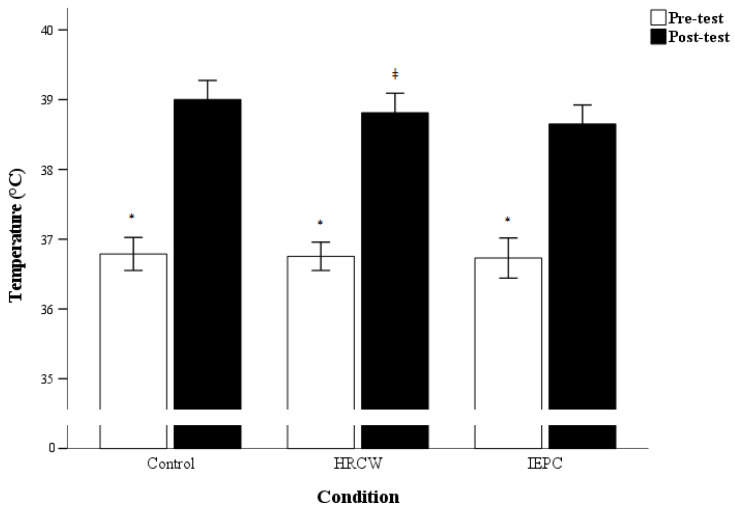
Combined effects of internal pre-cooling and hydrogen-rich water or external lower limb pre-cooling on body temperature before and after the shuttle run test under each condition. Note: Data are expressed as mean ± standard deviation; Control: internal pre-cooling; HRCW: hydrogen-rich cold water; IEPC: internal and external pre-cooling * significant pre-test difference; ǂ significant post-test difference between IEPC vs. control condition.

**Table 1 medicina-61-01173-t001:** Characteristics of the participants.

Participants	N	Age (Years)	Body Weight (kg)	Body Height (m)	BMI (kg.m^−2^)	Training (h wk^−1^)
Male sports science students	22	19.32 ± 0.84	59.82 ± 10.41	1.70 ± 0.11	20.59 ± 1.98	16

Notes: values are mean ± standard deviation; BMI: body mass index.

**Table 2 medicina-61-01173-t002:** Effects of internal pre-cooling intervention mixed with external pre-cooling using hydrogen-rich cold water on aerobic performance.

	Control	HRCW	IEPC	*p*	η^2^*_p_*	Power
MAS (Km/h)	12.24 ± 1.80	12.83 ± 1.78 ǂ	12.71 ± 1.83	0.009	0.464	0.857
Number of shuttle run repetitions	91.62 ± 45.19	110.29 ± 48.49 ǂ	104 ± 47.95	0.004	0.447	0.908
HRmean (beat/min)	168.90 ± 10.20	167.89 ± 10.38	165.91 ± 10.03	0.716	0.071	0.124
HRpeak (beat/min)	181.29 ± 9.65	181.38 ± 8.48	180.48 ± 7.19	0.222	0.211	0.351
Dehydration (%)	−0.27 ± 0.27	−0.44 ± 0.26 ǂ	−0.44 ± 0.19 *	0.016	0.429	0.799
Lac.post.test (mmol/L)	10.62 ± 0.82	9.98 ± 0.76 ǂ	10.01 ± 0.73 *	<0.001	0.771	1
RPEpost-test (AU)	9.67 ± 0.48	9.38 ± 0.50 ǂ	9.19 ± 0.60 *	0.036	0.370	0.685
FSpost-test (AU)	1.14 ± 1.56	2.38 ± 0.80 ǂ	2.24 ± 1.04 *	0.014	0.439	0.816

Note: Data are expressed as mean ± standard deviation; Control: internal pre-cooling; HRCW: hydrogen-rich cold water; IEPC: internal and external pre-cooling; MAS: maximal aerobic speed; RPE: rate of perceived exertion; HRmean: mean heart rate; HRpeak: peak heart rate; FS: feeling scale; Lac: lactate; η^2^*_p_*: partial eta squared; * significant difference between IEPC vs. control condition; ǂ significant difference between HRCW vs. control condition.

## Data Availability

The raw data supporting the conclusions of this article will be made available by the authors upon request.

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
