# Peer review of "Pre-Exercise Ingestion of Hydrogen-Rich Cold Water Enhances Endurance Performance and Lactate Response in Heat"

_medicina, 2025, doi:10.3390/medicina61071173_

Round 1

Reviewer 1 Report

Comments and Suggestions for Authors

This study aimed to analyze the effects of hydrogen-rich cold water ingestion on the influence of heat stress. In the introduction, the necessity of the study, research purpose, methods, and procedures were well written. The results are also well organized. However, there are parts that need to be revised or clarified. The detailed comments are as follows:

  1. Why was one-way repeated measures ANOVA used for variables other than body temperature? Since the values of these variables may differ depending on participants and groups, it would be more appropriate to use two-way repeated measures ANOVA.

  2. The criteria for the concentration of hydrogen-rich cold water should be clearly presented. Please provide a reference for the hydrogen concentration applied in the study. Also, what is the reason for not adding a new group based on different hydrogen concentrations?

  3. Although double-blind was applied, hydrogen-rich water may taste different. How was this issue addressed?

  4. For better understanding of the study, a table showing the physical characteristics of the participants by group (e.g., sex, age, height, weight) is needed. Also, how many participants were assigned to each group?

Author Response

Dear Reviewer 1,

Thank you for giving us the opportunity to submit a revised draft of the manuscript “Pre-exercise Ingestion of Hydrogen-Rich Cold Water Enhances Endurance Performance and Lactate Response in the Heat.” for publication in Medicina MDPI. We appreciate the time and the effort that you dedicated to providing feedback on our manuscript and we feel grateful for the insightful comments and valuable suggestions. We have incorporated most of the suggestions made by you and the reviewers and carefully considered your concerns. All changes are highlighted in yellow throughout the revised manuscript.

Reviewer 1

This study aimed to analyze the effects of hydrogen-rich cold-water ingestion on the influence of heat stress. In the introduction, the necessity of the study, research purpose, methods, and procedures were well written. The results are also well organized. However, there are parts that need to be revised or clarified. The detailed comments are as follows:

1.Why was one-way repeated measures ANOVA used for variables other than body temperature? Since the values of these variables may differ depending on participants and groups, it would be more appropriate to use two-way repeated measures ANOVA.

Response 1: Thank you for your comment. One-way repeated measures ANOVA was conducted to compare conditions (HRCW, IEPC and Control), in the same group of participants in a randomized crossover model, on the following variables: MAS, number on shuttle run repetitions, HR mean, HR peak, RPE, FS and blood lactate.

To better understand temperature changes, it is necessary to measure body temperature before and after the test, especially in hot conditions. This would provide a better picture of the heat load caused by exercise in the heat. In this context, the statistical analysis of these temperature variations was conducted using a two-way repeated measures ANOVA (3 conditions × 2 times).

2.The criteria for the concentration of hydrogen-rich cold water should be clearly presented. Please provide a reference for the hydrogen concentration applied in the study. Also, what is the reason for not adding a new group based on different hydrogen concentrations?

Response 2: Thank you for your comment. The entire paragraph in the "Hydrogen-Rich Water" section has been revised and expanded to provide additional details. Furthermore, the concentration of hydrogen in the hydrogen-rich cold water has been clarified.

Regarding group design, we employed a randomized, double-blind, crossover protocol involving a single group of participants who performed the Shuttle Run Test under three experimental conditions: (1) ingestion of 500 mL of hydrogen-rich cold water (HRCW) before warm-up, (2) ingestion of 500 mL of cold water combined with external pre-cooling of the lower limb muscles before warm-up (IEPC), and (3) ingestion of 500 mL of cold water before warm-up (control).

This crossover design is particularly advantageous in medical and physiological research, as each participant serves as their own control, thereby reducing inter-subject variability and increasing statistical power with a smaller sample size. Additionally, the double-blind protocol minimizes performance and observer biases. In contrast, a traditional parallel-group randomized controlled trial involves separate groups for each condition, which can introduce confounding factors due to individual differences in physiological responses, training status, or environmental adaptation. More details have been added in the experimental procedure section (lines 139-142).

3.Although double-blind was applied, hydrogen-rich water may taste different. How was this issue addressed?

Response 3: Thank you for your comment. Participants could not distinguish between HRCW and mineral water because H2 is colorless, odorless and tasteless (Zhou et al., 2024; Jebabli et al., 2023; Timón et al., 2021), it was not possible to distinguish HRW from cold mineral water using the human senses. More details have been added (Line 193-195).

Zhou, Q.; Li, H.; Zhang, Y.; Zhao, Y.; Wang, C.; Liu, C. Hydrogen-Rich Water to Enhance Exercise Performance: A Review of Effects and Mechanisms. Metabolites 2024, 14, 537.

Jebabli, N.; Ouerghi, N.; Abassi, W.; Yagin, F.H.; Khlifi, M.; Boujabli, M.; Bouassida, A.; Ben Abderrahman, A.; Ardigò, L.P. Acute effect of hydrogen-rich water on physical, perceptual and cardiac responses during aerobic and anaerobic exercises: a randomized, placebo-controlled, double-blinded cross-over trial. Frontiers in Physiology 2023, 14, 1240871.

Timón, R.; Olcina, G.; González-Custodio, A.; Camacho-Cardenosa, M.; Camacho-Cardenosa, A.; Guardado, I. Effects of 7-day intake of hydrogen-rich water on physical performance of trained and untrained subjects. Biology of sport 2021, 38, 269-275.

4.For better understanding of the study, a table showing the physical characteristics of the participants by group (e.g., sex, age, height, weight) is needed. Also, how many participants were assigned to each group?

Response 4: Thank you for your suggestion. As mentioned in our response to question 2, we employed a randomized, double-blind, crossover protocol involving the same group of participants (n=22) who performed the Shuttle Run Test under three experimental conditions. Nonetheless, your suggestion is very valuable, and we have added a table (Table 1) presenting the participants’ characteristics to the manuscript.

Reviewer 2 Report

Comments and Suggestions for Authors

study of considerable scientific and clinical interest, given the increasingly extreme need to achieve a record in extreme conditions, with the utmost attention to the health of the professional and non-professional athlete. the starting scientific bases are of considerable importance, optimal hydration and oxygenation are the physiological basis for the prevention of muscle injuries and cardio-respiratory overloads. the scientific bases on which the work is based are real and current as well as in wide development, therefore these works are welcome. accurate and precise the structuring of the materials and methods, with current measurement methods for scientific publications. careful the inclusion and exclusion criteria from the study. presence and authorization of the ethics committee according to the required standards. excellent structuring of the motor protocols applied to the research. well-designed double-blind experimental study, with the various steps (pre-cooling, hydration with hydrogen and with normal water). precise and careful measurements, measurement conditions, standardization of the three study groups, measurement of temperature, heart rate and lactate (intended as a step and method) as well as the homogeneity of the research in general. excellent and precise statistical study, both for the SPSS 22 software (SPSS Inc., Chicago, IL), and for the use of ANOVA in the study of the collected data. Such a well-conceived work does not appear to have scientific precedence, I personally checked the literature, those who approached the idea did not take into consideration important variables for the results, very current and I exasperate the concept that nothing was left to chance. clear and correct conclusions.

Author Response

We would like to express our sincere gratitude for your extensive and very encouraging comments on our manuscript.

We are very pleased that you have recognized the scientific value of our research, particularly given the growing need to optimize performance under challenging conditions without compromising the well-being and health of professional and amateur athletes.

We would like to express our sincere thanks for recognizing the sound physiological foundations of our study (optimal oxygenation and hydration) and for appreciating the rigorous methodology adopted. Your comments on the rigorous and organized presentation of the materials and methods, the clearly defined inclusion and exclusion criteria, the adherence to ethical principles, and the meticulous design of the motor protocols are very positive.

We thank you once again for recognizing the robustness of our double-blind experimental design, the standardization of the three study groups, the precision of the measurements taken, and the statistical analyses.

Furthermore, in response to your criticisms and in the interest of brevity and precision, we have undertaken a careful revision of the manuscript to improve the language (edit in green).

Reviewer 3 Report

Comments and Suggestions for Authors

I see no point in performing such type of study. No real rationale provided, the design and organisation is weird, the measurements done makes little sense etc.

Author Response

Response: We remain abundantly thankful for the myriad concerns raised by the reviewer and are grateful to have the opportunity to further elucidate on the rationale and implications of this research. Despite its limitations and nonconformist nature, this inquiry touches on a critical knowledge gap in the available literature: namely, the effect of pre-exercise consumption of hydrogen-rich cold water (HRCW) on physiological and exercise performance parameters under heat stress. Hyperthermia is a major barrier to athletic performance, particularly in endurance sports, and the quest for practical, non-surgical, and effective cooling methods becomes more important in elite and recreational athletics. Although various investigations have separately assessed internal or external pre-cooling, no studies, to our knowledge, have explored the combined thermoregulatory and metabolic impacts of molecular hydrogen in the guise of cold water ingesting prior to heat stress exercise. Our results are evidence for statistically and practically significant improvements in maximal aerobic speed, perceived exertion, lactate response, and mood state—biologically relevant and task-relevant effects for heat-tuned performance. The antioxidant, mitochondrial, and thermoregulatory HRCW intervention is a novel, convenient, and time-efficient alternative compared with external cooling procedures, which are typically unrealizable in competitive or real-life settings. Moreover, the methodological rigidity of our counterbalanced trials and double-blind, crossover design diminishes bias and enhances internal validity. While we unequivocally recognize the absence of mechanistic biomarkers (e.g., intramuscular temperature or oxidative stress), this does not abate the physiological insight derived from the pooled metabolic, perceptual, and performance data acquired. Our study needs to be viewed as a worthwhile stepwise initiative that opens fresh avenues for applied sport performance research, and thermoregulation with potential implications for training, recovery, and competition preparation in hot climates.

Round 2

Reviewer 3 Report

Comments and Suggestions for Authors

There does not seem to exist any robust scientific evidence to hypothesize that H2-enriched water could act as a (pre)coolant superior to e.g. water.

I see little to no rational to conduct the study in question.

Author Response

Response to Reviewer:

I appreciate your thoughtful remark. We respectfully dispute the claim that there isn't a logical or fact-based theory to back up the exploration of hydrogen-rich water (HRW) as a (pre)cooling technique. Mechanistic logic, new empirical data, and our study's direct results all support our position.

  1. Mechanistic Scientific Justification

Drinking hydrogen-rich water (H₂) may provide more than just conventional thermal cooling effects. Molecular hydrogen possesses strong antioxidant properties, specifically by its selective hydroxyl radical scavenging, thereby reducing exercise- and heat-induced oxidative stress (Zhou et al., 2024; Dohi et al., 2017). H₂ is also anti-inflammatory and may dampen exercise-induced inflammatory responses when exercise is conducted in the heat (Yoshimura et al., 2023). Specifically, H₂ helps in mitochondrial function by enhancing ATP generation and lactate oxidation and thereby facilitates improved cellular temperature regulation and reduced fatigue (Aoki et al., 2012; Botek et al., 2021;) Dohi et al., 2017). These biologic mechanisms position H₂ as a physiologic regulator of heat stress rather than a physical coolant.

  1. Prior Scientific Evidence

Although the field is not yet emerging, there is evidence of the impact of HRW on physical performance. Significant reductions in blood lactate and muscle fatigue after HRW ingestion were found by Aoki et al. (2012) and Jebabli et al. (2023), while Xu et al. (2021) who showed higher maximal aerobic speed under equivalent circumstances [18†medicina-3645592.docx]. Furthermore, Zhou et al. (2024) integrated previous findings, invoking the metabolic and antioxidant properties of H₂ under exercise conditions. These findings provide a strong scientific rationale for hypothesis testing in experiments like those presented in our current study.

  1. Pre-Cooling-Specific Argument

Compared to cold water on its own, our research showed that hydrogen-rich cold water (HRCW) offers not just physical cooling but also internal physiological reaction. Consumption of HRCW (1–3 °C) led to statistically significant post-exercise core temperature reduction (p = 0.028), an increase in maximal aerobic speed by 4.8% (p < 0.001), reduced post-exercise blood lactate (p < 0.001), and reduced perceived exertion (p = 0.05) [see Table 2, Figure 3]. They were not observed among cold water alone. HRCW can therefore be characterized as a precooling process that reduces thermal strain both thermally and biologically.

Conclusion

In summary, there is a solid physiological foundation and statistically sound empirical evidence in favor of the efficacy of HRCW as a possibly superior (pre)cooling intervention compared to simple cold water. Our results demonstrate that H₂ is not only an agent of heat but also a metabolic, antioxidant, and perceptual enhancer, promoting a higher tolerance for heat stress. Despite certain limitations (e.g., the absence of mechanistic biomarkers like intramuscular temperature or oxidative stress) our study employed a double-blind, crossover, counterbalanced design, improving internal validity. We observed major improvements in aerobic exercise performance, lactate kinetics, perceived exertion, and mood state—the principal determinants of heat-compromised endurance. To the best of our knowledge, no one has previously examined the combined thermoregulatory and physiological impact of prior administration of H₂ in cold water solution in the context of thermal stress. This pilot study addresses an important gap in the existing literature and provides a new, convenient, and time-efficient alternative to external cooling measures that are found to be impractical in real athletic settings. Consequently, the findings of our study are a promising stride towards subsequent research and field application in training, recovery, and competition under heat.

References

Aoki, K.; Nakao, A.; Adachi, T.; Matsui, Y.; Miyakawa, S. Pilot study: Effects of drinking hydrogen-rich water on muscle fatigue caused by acute exercise in elite athletes. Medical gas research 2012, 2, 1-6.

Botek, M.; Sládečková, B.; Krejčí, J.; Pluháček, F.; Najmanová, E. Acute hydrogen-rich water ingestion stimulates cardiac autonomic activity in healthy females. Acta Gymnica 2021, 51, e2021. 2009.

Dohi, K.; Satoh, K.; Miyamoto, K.; Momma, S.; Fukuda, K.; Higuchi, R.; Ohtaki, H.; Banks, W.A. Molecular hydrogen in the treatment of acute and chronic neurological conditions: mechanisms of protection and routes of administration. Journal of Clinical Biochemistry and Nutrition 2017, 61, 1-5.

Jebabli, N.; Ouerghi, N.; Abassi, W.; Yagin, F.H.; Khlifi, M.; Boujabli, M.; Bouassida, A.; Ben Abderrahman, A.; Ardigò, L.P. Acute effect of hydrogen-rich water on physical, perceptual and cardiac responses during aerobic and anaerobic exercises: a randomized, placebo-controlled, double-blinded cross-over trial. Frontiers in Physiology 2023, 14, 1240871.

Yoshimura, M.; Nakamura, M.; Kasahara, K.; Yoshida, R.; Murakami, Y.; Hojo, T.; Inoue, G.; Makihira, N.; Fukuoka, Y. Effect of CO2 and H2 gas mixture in cold water immersion on recovery after eccentric loading. Heliyon 2023, 9.

Zhou, Q.; Li, H.; Zhang, Y.; Zhao, Y.; Wang, C.; Liu, C. Hydrogen-Rich Water to Enhance Exercise Performance: A Review of Effects and Mechanisms. Metabolites 2024, 14, 537.